# Composite Membrane Containing Titania Nanofibers for Battery Separators Used in Lithium-Ion Batteries

**DOI:** 10.3390/membranes13050499

**Published:** 2023-05-08

**Authors:** Hun Lee, Deokwoo Lee

**Affiliations:** 1Applied Chemistry, Division of Energy & Optical Technology Convergence, College of Engineering, Cheongju University, Cheongju 28503, Republic of Korea; 2Department of Computer Engineering, Keimyung University, Daegu 42601, Republic of Korea

**Keywords:** electrospinning membrane, inorganic nanofiber, battery separator

## Abstract

In order to improve the electrochemical performance of lithium-ion batteries, a new kind of composite membrane made using inorganic nanofibers has been developed via electrospinning and the solvent-nonsolvent exchange process. The resultant membranes present free-standing and flexible properties and have a continuous network structure of inorganic nanofibers within polymer coatings. Results show that polymer-coated inorganic nanofiber membranes have better wettability and thermal stability than those of a commercial membrane separator. The presence of inorganic nanofibers in the polymer matrix enhances the electrochemical properties of battery separators. This results in lower interfacial resistance and higher ionic conductivity, leading to the good discharge capacity and cycling performance of battery cells assembled using polymer-coated inorganic nanofiber membranes. This provides a promising solution via which to improve conventional battery separators for the high performance of lithium-ion batteries.

## 1. Introduction

Rechargeable lithium-ion batteries have been widely studied and developed to satisfy high energy demands in various applications, such as portable electronic devices, hybrid electric vehicles, and a variety of power tools. Lithium-ion batteries provide good power storage due to their high energy density, excellent cycle life, and lack of memory effect [1].

The separator in lithium-ion batteries is the essential part in battery kinetics, preventing the physical contact of the positive and negative electrodes and maintaining battery performance. The separator must be mechanically and electrochemically stable with the electrolyte and electrode in lithium-ion batteries. Battery separators have many requirements, including wettability, thickness, pore size, porosity, thermal stability and safety [2,3]. A variety of membranes have been tested for use in commercial lithium-ion batteries over the last few decades. Most of the separators in lithium-ion batteries use microporous polyolefin membranes, including polyethylene, polypropylene, and their combinations. Although microporous polyolefin membranes provide excellent mechanical strength and good chemical stability, they exhibit a low thermal stability, which causes a short circuit between electrodes, and poor wettability to a polar liquid electrolyte, leading to high resistance and a low battery performance [4,5]. In order to overcome these drawbacks of polyolefin membranes, various approaches have been studied, including the surface modification of polyolefin membranes, the impregnation of a gel polymer electrolyte, nonwoven separators, and inorganic composite separators [6,7,8,9,10]. Among these, inorganic composite separators have received considerable attention due to their outstanding thermal stability and good wettability to a liquid electrolyte [11,12,13,14]. Adding inorganic materials promotes the migration of lithium ions in the electrolyte solution and reduces their crystallinity. The battery cells assembled using the resultant membranes have exhibited improved electrochemical performance due to the presence of inorganic materials. For instance, a composite membrane containing colloidal alumina was reported by Ali et al. [15]. They fabricated a blended membrane using colloidal Al_2_O_3_ via the phase inverse method. The resultant membrane exhibited a high electrolyte uptake, good ionic conductivity, and a stable cycling performance. Xu et al. [16] revealed that the membrane separator consisted of glass fiber fabrics with silica nanoparticles. It showed good mechanical strength and suppressing properties regarding the growth of lithium dendrite. Zhou et al. [17] prepared a novel composite separator by incorporating a silicon nitride whisker into a porous substrate via a phase separation method. It was mentioned that the inorganic whiskers in the resultant composite membrane contributed to high an electrolyte affinity and thermal stability, leading to good electrochemical performances. However, introducing inorganic particles is associated with challenges regarding the homogenous dispersion of nano particles in a polymer matrix.

Electrospinning technology is a novel and efficient method used to fabricate nanofiber-based membranes that have a high porosity and small pore size [18]. In addition, electrospun nanofibers can easily form an interconnected and uniform structure, which is an advantage in terms of combining both organic and inorganic materials. These nanofiber-based membranes have shown swollen nanofibers with increased fiber diameters after absorbing the electrolyte, resulting in a high electrolyte uptake and good electrochemical properties [19]. However, electrospun nanofiber membranes are typically weak and do not have sufficient mechanical properties to aid in the battery assembling process. The morphology and crystal structure of electrospun membranes were investigated by Gao et al. [20] in order to understand the electrospun membrane. Self-standing electrospun membranes have insufficient strength to withstand the penetration of electrode materials. If particular materials from the electrodes pass through the separator, this leads to electrical short circuit and the material can be easily damaged during the battery cycle. In order to overcome these drawbacks, a variety of approaches have been applied for the fabrication of membrane separators. One method that is particularly attractive regarding the enhancement of electrospun membranes involves introducing inorganic nanoparticles, such as TiO_2_ [21], SiO_2_ [22] and Al_2_O_3_ [23], into the polymer solution for electrospinning. For example, in one study, SiO_2_ was dispersed in an electrospinning solution with a PVDF polymer [22]. The resultant membranes exhibited an increment in their mechanical properties and an increase in their inorganic filler content. Zhai et al. [24] fabricated a multilayer separator consisting of PVDF/PMIA/PVDF by sequentially electrospinning different polymer solutions. The tri-layer separator showed better strength than the monolayer PVDF separator due to the existence of a PMIA layer, which had superior mechanical strength.

In addition to electrospinning, many researchers have studied other methods of using various polymers as polymer electrolytes, such as polyacrylonitrile (PAN), poly(vinyl chloride) (PVC), poly(methyl methacrylate) (PMMA), and poly(vinylidenefluoride) copolymer (PVDF-HFP, PVDF-CTFE). Fluoro-polymers have been widely used as a host material in the separators of lithium-ion batteries due to their good ionic conductivity and high electrochemical stability in an organic electrolyte [22,25,26,27,28,29,30,31]. For example, Cui et al. [32] developed a PVDF/polyester (PET) separator that was prepared via a thermally induced phase separation technique. The blend separator showed better thermostability and a higher capacity compared with the pristine membranes. A microporous cellulose/PVDF membrane was also prepared via the phase inversion method [33]. The novel separator resulted in high porosity, an increased electrolyte uptake, better ion conductivity, and a good electrochemical window due to its unique microstructure and polymer properties.

This paper presents the preparation and investigation of a novel composite membrane for use as a separator in lithium-ion batteries. The composite separator is based on the introduction of a ceramic nanofiber membrane, which is expected to improve its thermal stability and wettability in liquid electrolyte. To combine the advantages of electrospun nanofibers and inorganic materials, electrospinning and heat treatment are used to obtain the inorganic nanofiber membrane. After that, it is coated with a PVDF-HFP copolymer to achieve the appropriate mechanical strength to be used as a battery separator. The thermal stability, wettability, and microstructure of the prepared composite membrane are investigated. Consequently, the electrochemical properties of the new separator are evaluated to enable its application in lithium-ion batteries.

## 2. Materials and Methods

### 2.1. Material

Titanium isopropoxide (TIP, Aldrich) and polyvinylpyrrolidone (PVP, Mw = 1,300,000, Aldrich) were used as the precursor for the inorganic nanofibers. The mixture was prepared by adding 14 wt.% TIP into a 4 wt.% PVP solution in isopropanol. The mixture was stirred at room temperature to obtain a homogeneous solution. A polymer solution of 15 wt.% polyvinylidene fluoride-co-hexafluoroethylene (PVDF-co-HFP, Mw = 477,000, Arkema) in N,N-dimethylformamide (DMF, Aldrich) and acetone (7:3 by weight) was prepared so that polymer coating onto inorganic nanofibers could be performed.

### 2.2. Preparation of the Composite Separator

The preparation process of the polymer-coated inorganic nanofiber membrane is shown in Figure 1. The mixture was loaded into a plastic syringe and electrospun to form nanofibers using a flow rate of 4 mL∙h^−1^ at 20 kV. The nanofibers were collected on an aluminum foil that was placed away from the syringe needle at 15 cm. The as-spun nanofibers were dried at 120 °C for 6 h and then calcined at 550 °C for 1 h in order to obtain inorganic nanofibers. The prepared inorganic nanofibers were coated with a PVDF solution using a casting blade with a gap of 50 μm. After this, the composite membrane was immediately immersed into a water bath as a nonsolvent and dried at room temperature for 24 h to complete the solvent–nonsolvent exchange.

### 2.3. Measurement

The morphology of the composite membrane was evaluated using scanning electron microscopy (JEOL 6400F Field emission SEM at 5 kV, Tokyo, Japan). The samples used for SEM observation were coated with Au/Pd by a K-550X sputter coater to reduce charging.

The liquid electrolyte uptake capacities of the membranes were measured by soaking pre-weighed separator membrane samples for a fixed time at room temperature in a liquid electrolyte, which consisted of 1 M Lithium hexafluorophosphate (LiPF_6_) dissolved in 1:1:1 (by volume) ethylene carbonate (EC)/dimethylcarbonate (DMC)/ethylmethyl carbonate (EMC). The electrolyte was absorbed both on the surface and in the pores of the membranes. The excess electrolyte solution that adhered to the membrane surface was removed by gently wiping it with filter paper. The electrolyte uptake capacities of the membranes were determined using the following Equation (1):Uptake Capacity (mg∙cm^−2^) = (W_t_ − W_0_)/A(1)
where W_t_ is the weight of the electrolyte-immersed membrane, W_0_ is the weight of the dry membrane, and A is the immersed area of the test sample.

For the ionic conductivity measurement, membranes were completely soaked in the liquid electrolyte of 1 M LiPF_6_ in EC/DMC/EMC and then sandwiched between two stainless-steel plate electrodes. The ionic conductivities of the liquid electrolyte–soaked membranes were measured via electrochemical impedance spectroscopy (EIS) using a Potentiostat/Galvanostat/ZRA device (GAMRY Reference 600). The measurement was carried out at amplitude of 10 mV over a frequency range of 1 Hz to 1 MHz at a temperature range of 25 to 100 °C. The temperature was controlled in a temperature/humidity chamber (Ransco RTH-600-S). The conductivity (σ) was calculated using the following Equation (2):σ = d/(R_b_ × A)(2)
where d is the membrane thickness, R_b_ the bulk resistance of the liquid electrolyte–soaked membrane, and A the cross-sectional area of the membrane.

The interfacial resistance between the liquid electrolyte–soaked membranes and the lithium electrode was measured by EIS using the Potentiostat/Galvanostat/ZRA. The liquid electrolyte–soaked membranes were symmetrically sandwiched between two lithium electrodes. The measurement was performed over a frequency range of 0.01 Hz to 65 kHz under open-circuit conditions.

The electrochemical performance of the lithium/lithium iron phosphate (LiFePO_4_) cells containing the liquid electrolyte–soaked membranes was evaluated using an automatic battery tester (Arbin BT2000). The cells were prepared using the electrolyte–soaked membrane sandwiched between the lithium and LiFePO_4_ electrodes. Charge and discharge cycles were conducted in the potential window of 2.5–4.2 V at a 0.2 C rate.

## 3. Results and Discussion

The morphologies of the processed membranes, including the polymer nanofiber membrane, inorganic nanofiber membrane, and polymer-coated inorganic nanofiber membrane, are presented in the SEM images of Figure 2.

The electrospun nanofiber membrane shows an interconnected fibrous structure with average fiber diameters in the range of hundreds of nanometers (Figure 2a). As shown in Figure 2b, the inorganic nanofiber membranes prepared via heat treatment consistently maintained their three-dimensional network structure with fully interconnected nanofibers. Figure 2c shows the surface morphology of the PVDF polymer-coated membrane, including its inorganic nanofibers. It is clear that the inorganic nanofibers are completely combined with the PVDF coating and incorporated inside the composite membrane, forming one structure whose components cannot detach from each other. The composite membrane forms an interconnected structure and tortuous pores, which are beneficial to a high electrolyte uptake and prevent the growth of a dendritic formation during the charging and discharging process. The microporous structure of the composite membrane acts as excellent reservoir for liquid electrolyte, increasing the amount of electrolyte.

One of the most important roles of the battery separator is preventing the physical contact of the positive and negative electrodes, alongside dimensional stability. In addition, the membrane separator has to be strong and flexible enough to handle battery assembly [34,35,36]. Although electrospun nanofibers simply form a three-dimensional fibrous structure, this is typically weak and can be easily damaged during the assembling process of batteries. It has been reported that membranes consisting of electrospun nanofibers show a lower mechanical strength than those used in the polypropylene separator [37]. In addition, the inorganic nanofiber membranes prepared via the thermal treatment are inherently weak and brittle [38,39]. They have insufficient mechanical properties and are thus unsuitable to withstand the battery assembling process. As shown in Figure 3a, as-spun inorganic nanofibers are very thin and weak, which means that they have insufficient physical strength to allow for handling, even though they maintain a fibrous microstructure, as shown in Figure 2b. On the other hand, polymer-coated inorganic nanofiber membranes exhibit free-standing and flexible properties, as shown in Figure 3b. The coating of the PVDF polymer onto the inorganic nanofibers provides the mechanical properties required for the stresses inherent in the manufacturing process of batteries.

Electrolyte uptake is one of the most important properties used to indicate the battery separator’s wettability in the liquid electrolyte. The separator has to absorb and retain a sufficient amount of liquid electrolyte to achieve high ionic conductivity.

Figure 4 shows the electrolyte uptake capacity of the polymer-coated inorganic nanofiber membrane. For comparison, the electrolyte uptake capacities of the polypropylene membrane are also presented. The commercial PP membrane has poor wettability to polar liquid electrolytes due to its intrinsically nonpolar and hydrophobic nature, which restricts the performance and cycle life of lithium-ion batteries. It is observed that polymer-coated inorganic nanofiber membranes absorb liquid electrolyte rapidly after being immersed in the liquid electrolyte. Figure 4 shows that the uptake process is very fast and more than 90% of the uptake takes place in 60 s. Furthermore, the uptake capacities level off after a few minutes, which indicates that the membranes rapidly reach the saturation point in terms of electrolyte uptake. The membrane separator’s fast absorption of the liquid electrolyte facilitates the process of electrolyte wetting in the battery assembly. The composite membranes show a higher electrolyte uptake capacity than that of the polypropylene membrane due to the PVDF polymer coating and inorganic nanofibers. The good uptake capacity of the composite membrane is attributed to its good affinity for the electrolyte and fully interconnected microporous structure, which increases the retention of the liquid electrolyte. The PVDF polymer in the composite membrane strongly interacts with the organic electrolyte and the polar groups of PVDF. The electrolyte solution can be easily absorbed and encapsulated within the pore of the composite membrane. Moreover, such a structure is very beneficial to the prevention of the growth of dendritic lithium during the charge–discharge cycles of rechargeable batteries. In addition to the microstructure, the liquid electrolyte is also absorbed into the amorphous matrix of the PVDF polymer, leading to a swollen phase that acts as the transport channel of ions. It also improves the adhesion between the membrane separator and electrodes [40]. This demonstrates that these effects contribute to high ionic conductivity and a better battery performance.

Battery separators must not noticeably shrink and wrinkle when the temperature rises. The thermal stability of the battery separator is required for the safety of lithium-ion batteries, because the dimensional shrinkage of the separator that is caused by the high temperature during cycling at high current rates leads to the internal short circuit failures of batteries [41]. To investigate the thermal stability of separators, the membranes are observed and their dimensional change after heat treatment at 150 °C for 30 min is measured. Figure 5 shows photographs of the polypropylene membrane and polymer-coated inorganic nanofiber membrane before and after heat treatment. Zhou et al. [17] revealed that the pristine PVDF membrane exhibits a dimensional change at temperatures over 150 °C and becomes transparent. As shown in Figure 5, the shape of the polypropylene membrane changes from round to oval, but the polymer–coated inorganic nanofiber membrane exhibits no shrinkage after thermal treatment. The good thermal stability of the composite membrane can be attributed to the thermally stable TiO_2_ nanofibers and the heat resistance of the PVDF polymer coating. The inherent thermal resistance of the TiO_2_ nanofibers is believed to prevent the composite membrane from shrinking. Furthermore, it is widely known that membrane separators prepared using a PVDF polymer show stable thermal properties that allow for battery safety [3,42]. The results of the thermal test demonstrate that the composite membrane effectively maintains its dimensional shape and avoids a short circuit between the electrodes at an elevated temperature.

The electrochemical properties of the battery separator play a critical role in determining the battery performance. Impedance spectroscopy offers valuable information regarding cell resistance over a wide frequency range. The interfacial resistance between the liquid electrolyte–soaked membrane and the lithium electrode is evaluated by using impedance spectra. Figure 6 Shows the Nyquist plots for the polypropylene and PVDF polymer-coated inorganic nanofiber membrane in the range of 0.0 Hz to 65 kHz at room temperature. The real axis intercept of the Nyquist plot at a high frequency indicates the bulk resistance (R_b_) of the liquid electrolyte–soaked membrane. The diameter of the semicircle determines the charge transfer resistance (R_ct_), which is the interfacial resistance associated with the interfacial transfer of lithium ions between a liquid electrolyte–soaked membrane and a lithium electrode [43,44]. As shown in Figure 6a, the polymer-coated inorganic nanofiber membrane shows a lower interfacial resistance than that of the polypropylene membrane. The PVDF coating layer on the inorganic nanofibers improves the migration of lithium ions at the interface of the electrolyte–soaked membrane and electrode. The high mobility of lithium ions and their high amounts are a result of the superior electrolyte uptake of the composite membrane, which is related to high porosity and the affinity of the membrane to liquid electrolyte. Therefore, the polymer-coated inorganic nanofiber membrane exhibits a lower level of electrochemical impedance than that of the polypropylene membrane. This result is because the high electrolyte uptake of the composite membrane provides better transportation channels for lithium ions. The ions move through two-phase paths, which are the liquid phase of the electrolyte solution retained in micropores and the swollen phase of a polymer [45]. The large amount of liquid electrolyte and the expanded volume of the PVDF polymer allow the ions to easily migrate between electrodes, leading to lower resistance. The improved interfacial resistance also presents the better compatibility and lower reactivity of the electrolyte with lithium metal.

The performance of a lithium-ion battery is influenced by the interfacial properties between the lithium metal and electrolyte separator. Using lithium metal gives rise to the formation of a solid electrolyte interface (SEI) layer due to its thermodynamic instability when it contacts the electrolytes on the surface. Thus, the characterization of the interfacial stability of the electrodes plays a critical role in lithium-ion batteries. In order to study the interfacial stability of the lithium metal–electrolyte separator, the electrochemical impedance spectra with different storage times are presented in Figure 6b. The interfacial resistances of the membranes can be obtained by measuring the semicircles of the Nyquist plot at high frequencies. The interfacial resistances of the membranes increase as a function of the storage time. The increase in the interfacial resistance is attributed to the formation of a passive layer, which is created via a reaction between the electrolyte solvents and the lithium electrodes during storage. The interfacial resistances of the battery cells using the polymer-coated inorganic nanofiber membrane are depicted with the storage time in Figure 7. The interfacial resistance of the polymer-coated inorganic nanofiber membrane increases as the storage time increases, and stabilizes after 400 h. The stability after a long period of storage is ascribed the ability of inorganic metals to scavenge and catch solvent impurities. The high surface area of the inorganic nanofibers is able to advantageously trap the solvent impurities and prevent their accumulation at the interface. In addition, the highly porous structure of the composite membrane contributes to stabilizing the interfacial properties and the chemical reaction of the liquid electrolyte to form a passivation layer on the lithium metal after a certain time.

The ionic conductivities are obtained from the intersection of the Nyquist plots with the real axis at high frequency [46,47]. These are calculated using Equation (2) and are presented in Figure 8. It is observed that the ionic conductivities of the membranes increase as the temperature increases. The increment in the ionic conductivity is attributed to the expanded free volume of the polymer membrane and the higher ion mobility that occurs with a rise in temperature [48,49]. The ionic conductivities of the polymer-coated inorganic nanofiber membranes are higher than those of the polypropylene membrane at any given temperature. For example, the conductivities of the liquid electrolyte–soaked composite membrane at 25, 40, 60, 80, and 100 °C are 2.32 × 10^−4^, 3.11 × 10^−4^, 3.88 × 10^−4^, 5.37 × 10^−4^ and 5.54 × 10^−4^ S∙cm^−1^, respectively. On the other hand, the ionic conductivities of the liquid electrolyte–soaked polypropylene membranes are 4.56 × 10^−4^, 5.33 × 10^−4^, 7.08 × 10^−4^, 7.55 × 10^−4^ and 9.46 × 10^−4^ S∙cm^−1^. It has been reported that the ionic conductivity of a pristine PVDF membrane is 3.2 × 10^−4^ S∙cm^−1^ at room temperature, while composite membranes show improved ionic conductivities due to the presence of inorganic fillers [9]. Jeong et al. [49] proposed membranes based on the electrospun nanofiber type and a PVDF film type. The ionic conductivity of the film-type PVDF was 8.54 × 10^−5^ S∙cm^−1^ at 25 °C. As shown in Figure 8, the presence of the PVDF polymer coating leads to improved ionic conductivity because it increases the electrolyte uptake with as the wettability of the membrane increases. It is assumed that the existence of inorganic materials in the membrane is able to increase the ionic conductivity, since it depends on the ion mobility of the electrolytic solution. The lower interfacial resistance of the composite membrane leads to higher ionic conductivity as it makes lithium ions easily and effectively migrate between electrodes. In addition, the affinity of the PVDF polymer for the liquid electrolyte enables the electrolyte reservoir to maintain the full wetting of the membranes, leading to the better performance of the battery cell. The polarity of the PVDF coating offers interaction with the lithium ions and forms the electric double layer, which encourages ion transportation via the effects of electro–osmosis and electro–kinetic surface conduction [50].

In this study, the discharge capacities of the charge–discharge performance in Li/LiFePO_4_ cells with liquid electrolyte–soaked membranes were evaluated. LiFePO_4_, which is usually used as a cathode material, has a theoretical capacity of 167 mAh∙g^−1^ and an operating potential of 3.4 V. Figure 9a,b show the initial charge–discharge curves of the lithium-ion half cells assembled using the liquid electrolyte–soaked membranes. The curves present two flat plateaus between 3.2 and 3.6 V during the charge–discharge cycles, indicating the extraction and insertion reactions of lithium between LiFePO_4_ and FePO_4_. This flat plateau observed at around 3.4 V versus Li/Li^+^ at 0.2 C corresponds to the Fe^2+^/Fe^3+^ redox reaction that reflects the reversible charge–discharge cycling behavior of LiFePO_4_ as a cathode material. The discharge voltage plateaus of the cell consisting of composite membranes are maintained at constant, indicating the stable kinetics of the electrochemical process. Moreover, the cell based on the polymer-coated inorganic nanofiber membrane used for the separator exhibits a high discharge of 154 mAh∙g^−1^, which is an improved discharge capacity compared to the cell using a polypropylene membrane. For every cycle, the cells using the composite membrane separator are likely to have higher discharge capacities than those using polypropylene membranes. This is attributed to the highly porous structure of the composite membrane and its superior wettability, which arises from its good affinity for the electrolyte; this leads to its better ionic conductivity. The higher discharge value of the cell with the composite membrane results from the fast and easy lithium ion transportation between the electrodes, which indicates that there is lower interfacial resistance. It also demonstrates that TiO_2_ nanofibers improve the migration efficiency of lithium ions in the composite membranes between electrodes because inorganic material, i.e., TiO_2_, can capture the acidic impurity and trace amounts of moisture in the liquid electrolyte. As shown in Figure 9b, the higher Columbic efficiency observed during the third cycle shows that the electrochemical reaction of the electrode–electrolyte at the interface becomes stable as the cycles progress.

Figure 10 presents the discharge capacity of the cells assembled using the polymer-coated inorganic nanofiber membrane. The discharge capacities of the cell using the composite membrane are higher than those of the cell using the polypropylene membrane separator by around 140 mAh·g^−1^. Furthermore, the cells made using the polymer-coated inorganic nanofiber membrane exhibit a good retention of the discharge capacity; this is 150 mAh∙g^−1^ at the 50th cycle, which is 90% of the theoretical capacity of LiFePO_4_ (167 mAh·g^−1^). This is ascribed to the good wettability and affinity of the composite membrane for the electrolyte, which lead to a higher uptake capacity and better ionic conductivity. Inorganic materials in the form of a fibrous network and microporous structure also enhance the electrochemical reaction at the interface between the electrode and electrolyte, resulting in a better cycle performance. It was observed that the cell using the pristine PVDF-HFP separator resulted in a reduced capacity fade over 20 cycles at a 0.2 C rate [17,51]. In addition, membranes made using inorganic powder showed unstable capacities due to the presence of agglomerated nano particles ruining the structural integrity of the membrane. However, the cell with the composite membrane containing inorganic materials that formed a web-like shape showed an improved discharge capacity and had stable discharge capacities during cycling. Unlike nanoparticles, nanofibers are homogeneously distributed in the PVDF coating. Inorganic nanofibers evenly dispersed in the composite membrane help to improve the stable discharge capacities of battery cycles. Moreover, inorganic materials and the highly porous microstructure of the composite membrane give rise to the suppression of lithium dendrite formation and growth, leading to the stable cycling performance of batteries. Therefore, the introduction of a PVDF-HFP coating and inorganic nanofibers into the membrane separator helps to improve the cycle performance of cells.

## 4. Conclusions

In this work, nanofiber membranes are prepared by electrospinning a solution of the prepolymer and precursor. The non-woven mat of the electrospun nanofibers is thermally treated and processed via solvent casting. The composite membranes made with inorganic nanofibers are free-standing and flexible, thus allowing them to handle the assembly process, and show an independently connected network of inorganic nanofibers within the polymer matrix. The electrolyte uptake capacities of polymer-coated inorganic nanofiber membranes increase due to the presence of inorganic nanofibers within polymer coatings and the polymer’s affinity for the electrolyte. Furthermore, the composite membranes result in improved ionic transportation and interfacial resistance. Hence, they present higher ionic conductivities than the commercial porous separator because the composite membrane has better wettability and a lower level of electrochemical impedance. The battery cells using the composite membrane separators exhibit better charge–discharge capacities and cycling performances compared to those using the polypropylene membrane, since using the composite membrane leads to enhanced ionic conductivity. These results obviously indicate that membranes made with inorganic nanofibers could potentially be applied in the battery separators of lithium-ion batteries.

## Figures and Tables

**Figure 1 membranes-13-00499-f001:**
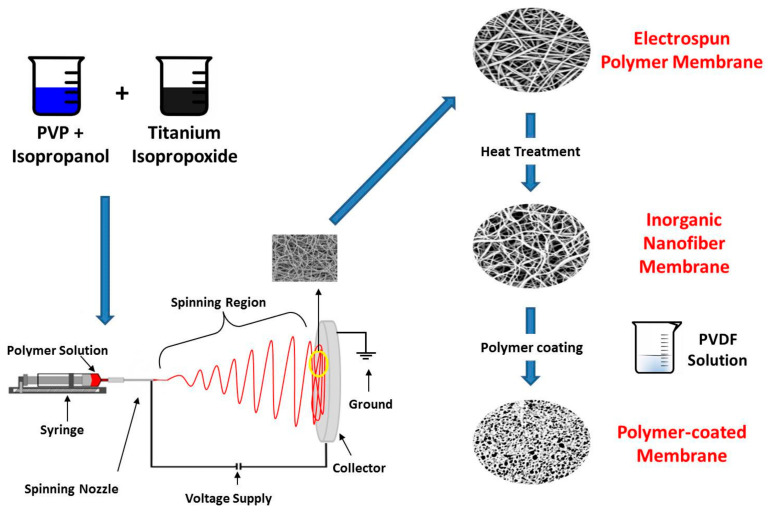
Schematic diagram of membrane preparation.

**Figure 2 membranes-13-00499-f002:**
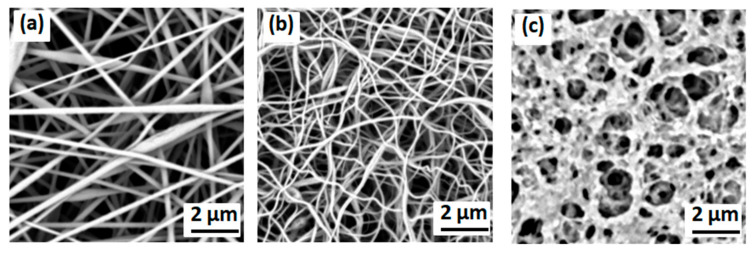
SEM images of (**a**) polymer nanofiber membrane, (**b**) inorganic nanofiber membrane, and (**c**) polymer-coated inorganic nanofiber(PCIN) membrane. Magnification: 20,000×.

**Figure 3 membranes-13-00499-f003:**
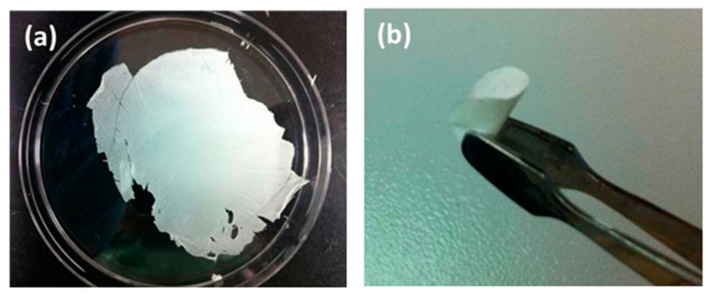
Photographs of (**a**) weak and brittle inorganic nanofiber membrane; (**b**) free-standing and flexible polymer-coated inorganic nanofiber membrane.

**Figure 4 membranes-13-00499-f004:**
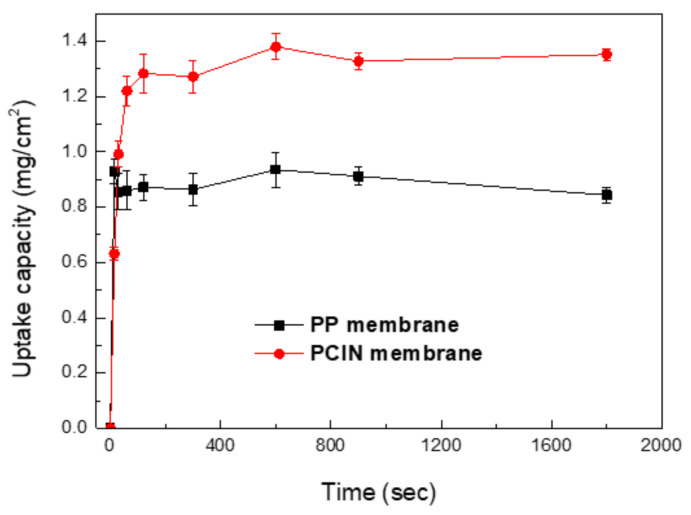
Electrolyte uptake capacities of polypropylene membrane and polymer-coated inorganic nanofiber membrane.

**Figure 5 membranes-13-00499-f005:**
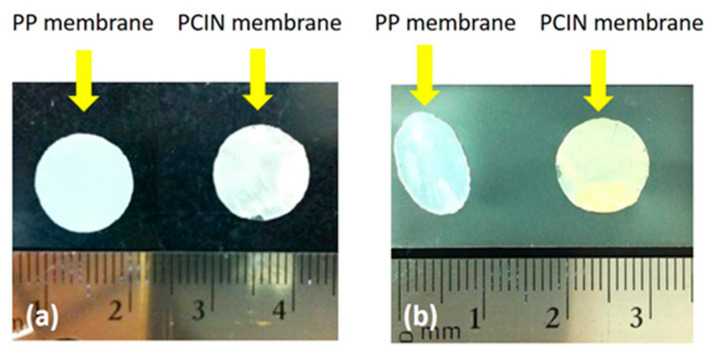
Photographs of polypropylene membrane (**left**) and polymer-coated inorganic nanofiber membrane (**right**) (**a**) before and (**b**) after thermal test at 150 °C for 30 min.

**Figure 6 membranes-13-00499-f006:**
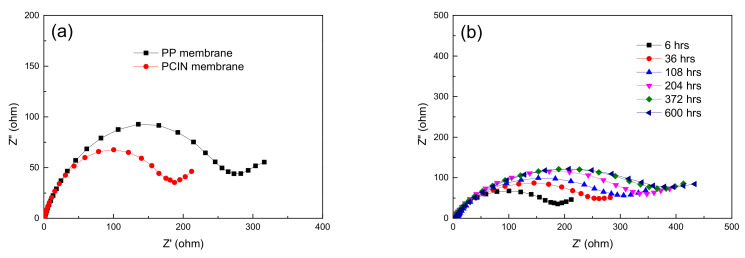
Electrochemical impedance spectra of the (**a**) polypropylene membrane and polymer-coated inorganic nanofiber membrane; (**b**) polymer-coated inorganic nanofiber membrane with various storage times at 25 °C.

**Figure 7 membranes-13-00499-f007:**
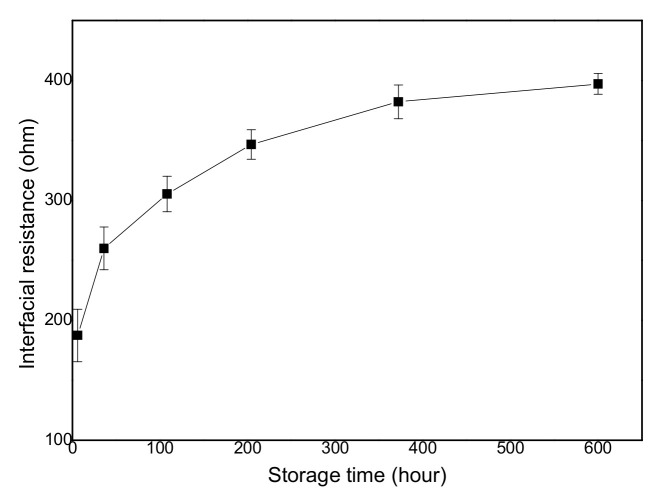
Variation in interfacial resistance of polymer-coated inorganic nanofiber membranes with different storage times at 25 °C.

**Figure 8 membranes-13-00499-f008:**
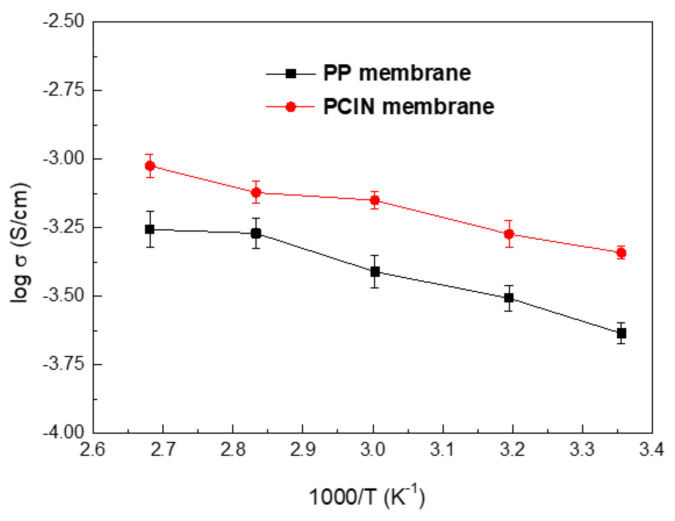
Ionic conductivities of liquid electrolyte–soaked membrane at various temperatures.

**Figure 9 membranes-13-00499-f009:**
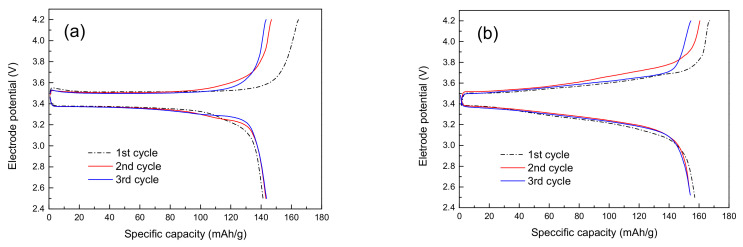
Charge–discharge curves of (**a**) polypropylene membrane and (**b**) polymer-coated inorganic nanofiber membrane at 0.2 C rate.

**Figure 10 membranes-13-00499-f010:**
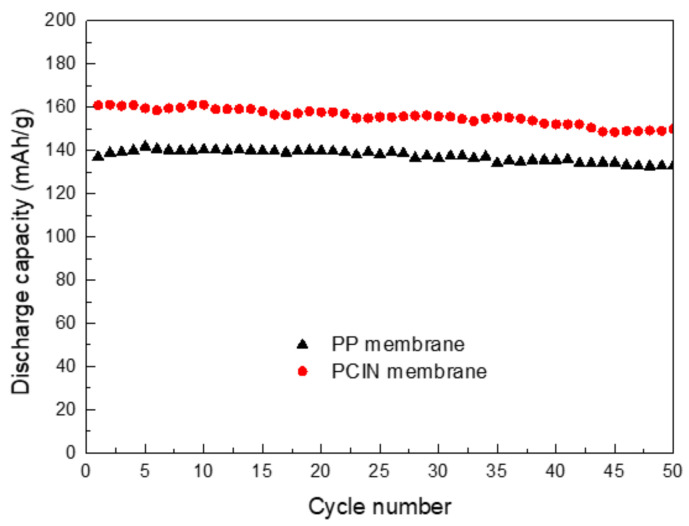
Cycle performance of polypropylene membrane and polymer-coated inorganic nanofiber membrane at 0.2 C rate.

## Data Availability

All data are included in the article.

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
