# Peer review of "Composite Membrane Containing Titania Nanofibers for Battery Separators Used in Lithium-Ion Batteries"

_membranes, 2023, doi:10.3390/membranes13050499_

Round 1

Reviewer 1 Report

The manuscript, "Composite Membrane Containing Titania Nanofibers for Battery Separators Used in Lithium-ion Batteries" has good results. The abstract and the introduction are very clear and reflect the aim of the work. The paper is well-organized and written. The methodology is quite good and clear for the readers. The authors succeeded in interpreting the results and the clear and sufficient conclusions. Thus, the paper is accepted for oublicatin.

The paper is acceptable for publication in Memebranes.

Reviewer 2 Report

Manuscript details the preparation of electro-spun nanofibers impregnated with an inorganic phase to improve the thermal stability and rigidity of electrode membrane in Li/LiFePO4 cell.

Manuscript is well designed and thought out. 

Fig 6. is missing so I could not verify the values given in the text. Authors should add this image.

" After this, the composite membrane was immediately immersed in the nonsolvent and dried at room temperature to complete the solvent-nonsolvent exchange." 

Give what is used as a nonsolvent.

English is good.

Only one thing should be amended "solution using by casting blade". Here "by" is not necessary, it should state "solution using casting blade"

English is good.

Only one thing should be amended "solution using by casting blade". Here "by" is not necessary, it should state "solution using casting blade"

Reviewer 3 Report

Figures 4 and 6 are missing.

Please add error bar for Figures 7 and 8

Any specific purpose on using  PVDF as a support for nanofiber?

Any comparison result for just using PVDF support (without Titania Nanofibers)? 

Round 2

Reviewer 3 Report

The paper can be accepted as it is.